# Towards a High Rejection Desalination Membrane: The Confined Growth of Polyamide Nanofilm Induced by Alkyl-Capped Graphene Oxide

**DOI:** 10.3390/membranes11070488

**Published:** 2021-06-29

**Authors:** Biqin Wu, Na Zhang, Mengling Zhang, Shuhao Wang, Xiaoxiao Song, Yong Zhou, Saren Qi, Congjie Gao

**Affiliations:** 1Center for Membrane and Water Science & Technology, Zhejiang University of Technology, Hangzhou 310014, China; Peachy_Wu@163.com (B.W.); zna9686@foxmail.com (N.Z.); zhangmengling07@outlook.com (M.Z.); shuhao100e@126.com (S.W.); gaocj@zjut.edu.cn (C.G.); 2Suzhou Institute of Nano-Tech and Nano-Bionics (SINANO), Chinese Academy of Sciences, Suzhou 215123, China

**Keywords:** two-dimensional nanomaterials, confined growth, diffusion control, interfacial polymerization, reverse osmosis membrane

## Abstract

In this paper, we used an octadecylamine functionalized graphene oxide (ODA@GO) to induce the confined growth of a polyamide nanofilm in the organic and aqueous phase during interfacial polymerization (IP). The ODA@GO, fully dispersed in the organic phase, was applied as a physical barrier to confine the amine diffusion and therefore limiting the IP reaction close to the interface. The morphology and crosslinking degree of the PA nanofilm could be controlled by doping different amounts of ODA@GO (therefore adjusting the diffusion resistance). At standard seawater desalination conditions (32,000 ppm NaCl, ~55 bar), the flux of the resultant thin film nanocomposite (TFN) membrane reached 59.6 L m^−2^ h^−1^, which was approximately 17% more than the virgin TFC membrane. Meanwhile, the optimal salt rejection at seawater conditions (i.e., 32,000 ppm NaCl) achieved 99.6%. Concurrently, the boron rejection rate was also elevated by 13.3% compared with the TFC membrane without confined growth.

## 1. Introduction

Polyamide thin-film-composite (TFC) reverse osmosis (RO) membranes have been widely used in desalination because of their low energy consumption and high separation efficiency [1,2]. The pursuit of high rejection and high selectivity RO membranes is one of the development areas that could yield high-quality product water and a more cost-effective process [3,4]. Secondly, the boron removal of the membrane needs to be further improved [5,6]. The selective layers in TFC-RO membranes are fabricated by the interfacial polymerization (IP) process, in which a polyamide (PA) film is formed at the interface of aqueous amine solution and organic acyl chloride solution. By doping nanomaterials into a PA matrix during interfacial polymerization (IP), researchers have fabricated various thin-film-nanocomposite (TFN) RO membranes. A group of nanomaterials can be exploited for this purpose. Some examples include: zeolites [7], carbon nanotubes (CNTs) [8], polyhedral oligomeric silsesquioxane (POSS) [9], graphene oxide (GO) [10], metal-organic frameworks (MOFs) [11], silica nanoparticles, etc. [12].

In recent years, the design and preparation of TFN membranes by combining new inorganic or organic nanomaterials with a traditional polyamide layer is a new research direction in the membrane separation field [13,14]. While most of these efforts are directed towards elevating the performance of the TFN membranes, the effect of the nanomaterials on the growth of the PA nanofilm, especially on the physicochemical aspects of the resultant PA nanofilm, has yet to be explicitly studied. According to recent studies [15,16,17], the intrinsic separation layer of the polyamide layer is the topmost nanofilm that forms the walls of nodular and leaf-like nanostructures on the polyamide membrane surface. Therefore, the thickness, surface area, and cross-linking degree of the intrinsic separation layer are closely related to the performance of the TFN membrane. Therefore, knowledge of the impact of nanomaterials on polyamide nanofilm formation is the key to well-constructed, high-performance TFN membranes.

Emerging studies have provided evidence that the growth of the polyamide nanofilm takes place in the organic phase [18,19,20], therefore blending nanomaterials in the organic phase should impact the growth of PA nanofilm by causing resistance to m-phenylene diamine (MPD) diffusion into the organic phase. Indeed, studies have shown that blending PIB (polyisobutylene) [21], zeolites [22], or MOFs (metal-organic frameworks) [11] could benefit the separation performance of the TFN membranes. Later, Yan et al. [23]. discovered that the ZIF-8 nanoparticles in the organic phase may confine the growth of the PA layer and therefore produce a PA layer with reduced apparent thickness. In this work, to further explore the barrier effect of nanomaterials, we have modified single-layer graphene oxide (GO) nanosheets with octadecylamine (ODA) to disperse the ODA@GO nanosheets into the organic phase. In this design, the MPD diffusion is largely confined to the molecular level because of the large lateral dimension of the ODA@GO nanosheets, which should enrich the MPD concentration at the organic/aqueous interface. Also, because of the macromolecular nature of the ODA@GO nanosheets, they will hardly diffuse and remain at the interface until the IP reaction terminates. Hence, this works aims to find out how the ODA@GO nanosheets impact the IP reaction and therefore the nanoscale structure and properties of the polyamide nanofilm. Further, we carried out the seawater desalination tests to reveal how these impacts are related to the performance of the resultant ODA@GO TFN membrane.

## 2. Experiments and Methods

### 2.1. Materials

Single layer graphene oxide powder (GO) was purchased from Hangzhou Gaoxi Technology Co., Ltd., Hangzhou, China. Octadecylamine (ODA, 97%), m-phenylene diamine (MPD, 99%), camphorsulfonic acid (CSA, 99%), triethylamine (TEA), 1,3,5-benzenetricarbonyl trichloride (TMC, 98%) and boric acid were purchased from Shanghai Aladdin Reagent Co. Ltd, Shanghai, China and used as received. Polysulfone (PSF) substrate membranes with a MWCO of 35 kDa were used from Huzhou laboratory pilot line, and deionized (DI) water with the electrical conductivity 1.6–2.3 was taken from the laboratory. Isopar-G was obtained from ExxonMobil Chemical Company, while n-hexane was from Shanghai Lingfeng Chemical Reagent CO., Ltd., Shanghai, China. Dehydrated alcohol (EtOH) was obtained from Anhui Ante Food Co., Ltd., Anhui, China and dimethylformamide (DMF) was purchased from Wuxi Haishuo Biological CO., Ltd., Wuxi, China. Sodium hydroxide (NaOH) and sodium chloride (NaCl) were purchased from Xilong Scientific Co., Ltd., Xilong, China and Guangdong Guanghua Sci-Tech CO., Ltd., Guangdong, China, respectively. All reagents were analytical grade unless otherwise stated.

### 2.2. Preparation of ODA@GO

Functional GO nanosheets were formed by binding octadecylamine (ODA) with oxygen-containing groups on GO, which can be seen from Figure 1. Briefly, 100 mg GO was dispersed in 50 mL DI water by bath ultrasound for 1 h. ODA solution (100 mg in 10 mL EtOH) was added into GO suspension and stirred well to blend. The mixed solution was poured into the 100 mL hydrothermal reactor and reacted at 90 °C for 24 h in a constant temperature oven. After the reaction, the resultant composite was rinsed with ethanol several times to remove unreacted ODA, then it was vacuum dried at 50 °C for 24 h [24]. The obtained black powder was stored for further usage and named as ODA@GO.

### 2.3. Characterizations of GO and ODA@GO

The Morphology of GO and ODA@GO was observed by field emission scan electron microscope (FESEM, SU8010, Hitachi) and atomic force microscopy (AFM, Bruker, Dimension Icon). First, a few drops of GO aqueous dispersion and ODA@GO hexane solution were dropped on a silicon wafer, to observe their morphologies by SEM. GO aqueous dispersion and ODA@GO hexane solution were also dropped on the mica wafer, to test their sizes and thickness by AFM.

The chemical compositions of the membranes were analyzed by Fourier transform infrared (FT-IR, ThermoFisher Nicolet-is50, USA) spectroscopy, X-ray diffraction (XRD, Panalytical-X’Pert Pro, Holland) and X-ray photoelectron spectroscopy (XPS, Kratos AXIS Ultra DLD, UK). GO, ODA and ODA@GO powder were mixed with KBr (mass ratio was 1:200) and compressed into a tablet to test the transmittance at room temperature by FTIR. The crystal structure of GO and ODA@GO nanosheets were detected by XRD with Cu Kα excitation radiation. The component element of GO and ODA@GO was analyzed by XPS using Al Kα (1486.6 eV) as the radiation source.

### 2.4. Preparation of RO Membrane

The TFC RO membranes were fabricated by the IP process, wherein 2.2% (*w*/*v*) MPD aqueous phase with CSA and TEA buffer solution (adjusted pH = 10) was reacted with 0.11% (*w*/*v*) TMC dissolved in isopar-G. The PSF ultrafiltration porous substrate was soaked in the MPD solution for 2 min, after which the residue was removed and then it was dried with sweeping N_2_. Subsequently, the TMC solution was impregnated for 1 min to remove the excess organic solution and form a thin layer. Finally, it was heated in an oven at 95 °C for 8 min to form a dense layer of PA that was named the virgin reference group.

The TFN RO membranes were prepared using the same steps above, but a series of ODA@GO nanosheets with different mass concentrations (0.001%, 0.003%, 0.005%, 0.01%, and 0.02% (*w*/*v*)) were added to the organic solution and mixed under bath ultrasonication for 1 h before the IP reaction. The sheets act as barriers in the growth process of PA. The prepared TFN membranes were named TFN-1 to TFN-5, in which a series of concentrations (from 0.001% to 0.02% (*w*/*v*)) of ODA@GO sheets were doped, respectively.

### 2.5. Characterization of RO Membrane

The fabricated RO membranes were cleaned with DI water and dried in a vacuum oven at room temperature for 24 h before the analyses were conducted. The top surface and cross-section of each membrane were examined by FESEM to observe the cross-sectional morphology. The samples were frozen in liquid nitrogen and then fractured. Before observation, all samples were coated with gold for 60 s. Transmission electron microscopy (TEM, HT7700, Hitachi) was conducted at 100 kV to examine the top surface and cross-section for further observing the morphology and evaluating the apparent and intrinsic thickness of the membranes. Briefly, after being separated from the PSF layer via DMF, the PA layer was overlaid on the top surfaces of copper grids thereafter to observe a specific morphology. The cross-sectional samples were embedded in resin for 8 h and then cut into approximately 80 nm-thick sections to place on the copper grids, respectively. AFM was used to observe the surface roughness of each 5 × 5 μm^2^ membrane by comparing Ra values from the obtained three-dimensional morphology images.

XPS was used to analyze the elemental content of the PA top surface within 10 nm of the PA layer by utilizing Al Kα (1486.6 eV) as the radiation source.

The hydrophilicity/hydrophobicity of each membrane surface was measured with a contact angle meter (CA, OCA15EC, Germany) using the sessile drop technique with DI water as the reference liquid. A droplet of DI water of approximately 3 μL was deposited on the leveled membrane surface to measure the contact angle of each sample. The mean static contact angle was calculated from six different positions.

A solid-surface zeta potentiometer (Zeta potential, Anton Paar SurPASS 3, Austria) was used to characterize the charge on the membrane surface over the pH range of 3–10. The background electrolyte solution was 1 mmol L^−1^ KCl. The pH was adjusted with 0.05 mol L^−1^ HCl and 0.05 mol L^−1^ NaOH.

### 2.6. Performance of the RO Membrane

The high-pressure cross-flow RO evaluation setup (Figure 2) was used to test the separation performance of the prepared membrane under brackish water and seawater conditions. In the process of our experiment, the flow rate and surface cross-flow velocity are 3 L/min and 0.31 m/s, respectively. Before the experiment, the device was operated for 1 h to stabilize the system pressure. First, the system was operated with pure water to calculate the water permeability coefficient (*A* value). Then, the system was operated with brackish water (2000 ppm NaCl solution) to calculate the solute permeability coefficient (*B* value) and with seawater water (32,000 ppm NaCl solution) to calculate the solute permeability coefficient (*B*′ value). The testing pressure was 16 bar for pure and brackish water testing and 55 bar for seawater. The other test conditions were constant (pH = 8; 25 °C). The *A* value (L m^−2^ h^−1^ bar^−1^) was calculated according to the equation *A* = *J*/ΔP, where ΔP is the operating pressure (bar); and *J* (L m^−2^ h^−1^) (LMH) is the permeate water flux, which is calculated according to the equation *J* = V/(A × ∆t), where V (L) is the volume of the permeate solution, A is the effective membrane area (19.63 cm^2^ for a single cycle module) and Δt (h) is the permeation time of the experiment. The *B* and *B*′ value (LMH) is calculated utilizing the equation *B* = *J*·(1 − *R*)/*R*, in which the rejection (*R* (%)) was calculated using the equation *R* = (1 − C_p_/C_f_)·100%, where C_p_ and C_f_ denote the concentrations of the permeate solution and feed solution, respectively. To further analyze the transport process of boron, the permeability (*B_s_*) of boron was calculated using the equation *J_s_* = *B_s_*·ΔC_s_, where *J_s_* is the permeation flux of the solute (NaCl), and ΔC_s_ is the concentration difference between the feed and permeate solution. The NaCl concentration was evaluated by conductivity meter (DDSJ-308A, Shanghai, China), whereas the boron concentration was measured using an inductively coupled plasma-optical emission spectrometer (ICP-OES, PerkinElmer, AvioTM 200).

## 3. Results and Discussion

### 3.1. Characterizations of GO and ODA@GO

The physiochemical properties of pristine GO and ODA@GO were characterized by FESEM, AFM, FTIR, XRD, and XPS (See Figure 3 and Figure 4). Digital photos of GO dispersion in water and isopar-G and ODA@GO in both solvents are listed from left to right (Figure 4a). The size of ODA@GO nanosheets was mainly around 1–2 um (Figure 3a,b) [25,26], which was consistent with the measurements of the AFM images (Figure 3c,d). The sheet size was calculated by line analysis using Nanoscope software. Meanwhile, the thickness of a single-layered ODA@GO increased to ~2.7 nm due to alkylation as compared to the virgin GO nanosheets (~1.3 nm). As shown in Figure 4a, unmodified GO was super-hydrophilic and dispersed in water instantly and was immiscible with isopar-G. While the modified hydrophobic ODA@GO was dispersed easily in isopar-G and not dispersible in the aqueous phase. Such oleophilic and hydrophobic properties render the ODA@GO nanosheets an ideal medium to inhibit the diffusion of MPD into the organic phase and hence confine the growth of the PA nanofilm, which is largely dependent on MPD diffusion [18,19,20].

The chemical compositions of GO and ODA@GO were analyzed using FTIR and XPS. The FTIR absorption spectra of the GO, ODA, and the ODA@GO were compared in Figure 4b. In the GO spectrum, bands were observed at 3397, 1719, 1637, 1100, and 683 cm^−1^, which are associated with the stretching vibrations of -OH, C=O, C=C, C-O, and C-H, respectively. In the ODA@GO spectrum, the peaks at 2920 cm^−1^, 2850 cm^−1^, and 721 cm^−1^ are attributed to ODA molecules assigning to C-H stretching, while characteristic peaks of C-N and N-H at 1467 cm^−1^ and 1577 cm^−1^ indicate that the epoxy group in the GO layer experienced a ring-opening reaction due to the nucleophilic substitution of protonated amino groups [27,28]. The XRD results of Figure 4c showed a diffraction peak at 2θ = 9° for the GO nanosheets, while the peak value of ODA@GO decreased to 2θ = 3.3°, which can be explained by the increment of the nanosheet spacing by intercalated alkyl chains. Meanwhile, a new weak peak of ODA@GO appeared at 21°, which might be related to the partial reduction of GO by ODA molecules [26,29]. Figure 4d,e showed the XPS analysis results of GO and ODA@GO, respectively. Comparing the C1s peak of GO and ODA@GO, it can be seen that the C-O peak fraction decreased significantly, and a new fraction attributed to C-N appeared at the same time, indicating that the epoxy group in the GO went through a ring-opening reaction [30], which agrees well with the FTIR results in Figure 4b. The above results confirmed the successful modification of GO from the physical and chemical aspects and the ODA@GO increased single layer thickness due to the grafting of ODA molecules onto the GO nanosheets.

### 3.2. Characterizations of TFC and ODA@GO TFN Membrane

#### 3.2.1. Morphology

The surface morphologies of the TFC and TFN membranes analyzed by SEM and TEM were shown in Figure 5a–c and Figure 5g–i respectively. The surface of the TFC membrane was characterized predominantly by a nodular structure. Then, the leaf-like structures, which are collapsed large-sized nodules [15], gradually appeared as the doping of ODA@GO increased. From the cross-sectional morphology of SEM Figure 5d–f of the RO membrane, we can observe that the PA nodular structure showed interconnected hollow voids inside, with the size of the majority of them less than 50 nm [19], which agreed well with the cross-sectional TEM images in Figure 5j–l. With more ODA@GO nanosheets doped in the organic solution, the apparent thickness of the PA layer (namely, the overall thickness of the PA layer) generally decreased from ~114.9 nm to ~69.2 nm (i.e., virgin membrane and the TFN-5 membrane, respectively. See Figure 6 and Table 1). In contrast to the apparent thickness, however, the intrinsic thickness (namely, the thickness of the polyamide nanofilm that forms the wall of the voids) increased from ~15.93 nm (for the virgin membrane) to ~21.19 nm (for the TFN-5 membrane). Interestingly, the pure water permeability coefficient *A* value of the TFN membranes increased gradually, which could be explained by the enhanced leaf-like structures on the former membrane favoring the higher water transportation surface area of the TFN membrane. Under the seawater condition performance test, the *B*′ values of TFN membranes decreased compared with the virgin TFC membrane, which was consistent with the increase in the membrane intrinsic thickness and cross-linking degree (see Figure 6 and Table 2). At the same time, the *A*/*B* value, which represents the selectivity of solvent (water) over the solute (NaCl), increased from 23.9 bar^−1^ (TFC membrane) to 26.63 bar^−1^ (TFN-4 membrane), pronouncing the optimizable selectivity of water and salt.

AFM was used to further explore the surface roughness of membranes (Table 1). The *R_a_* value of the TFC membrane was ~47.3 nm. After adding ODA@GO into the TMC solution, the TFN membranes became relatively smoother and the roughness decreased to ~33.5 nm (for 0.02% (*w*/*v*) loading). This is because the nodule characterized surface of the TFC membrane generally transfers into the leaf-like structure characterized membrane surface of the TFN membranes. These leaf-like structures are essentially collapsed large nanobubbles in the dry state. They overlap with each other and conceal the roughness beneath their flat structures, therefore reducing the surface roughness of the membrane [15,17]. We will address this phenomenon more systematically by combining other experimental details in Section 3.3 after addressing the chemical aspects.

#### 3.2.2. Chemical Analysis

The element composition of the membrane surface was determined by XPS (Figure 7). Compared to the TFC membrane, the C-C/C=C main peak area observed on the TFN membrane at 284.2 eV was increased, which should be attributed to the alkyl chains in ODA@GO. With the increase of the doping amount, two other main peak areas at 285.6 eV (C-O) and 284.8 eV (C-N) increased progressively, which should be attributed to the ring-opening of epoxy group in GO and the addition of amine group in ODA [26]. At the same time, as the elemental composition showed in Table 2, the greater the ODA@GO addition, the higher the C content was, and the O/C ratio decreased accordingly, which indicates a more hydrophobic membrane top surface [31]. Also, such analysis was further supported by the higher water contact angle for the ODA@GO incorporated membranes. Specifically, the contact angle increased from 84° for the virgin up to 134° for the TFN-5 membrane (Figure 8a). These analyses collectively suggest that ODA@GO nanosheets are partially incorporated in the top surface of the TFN membranes.

Theoretically, a fully cross-linked polyamide layer should exhibit an O/N value of ~1, and is only linearly-linked with an O/N ratio of 2 [32]. Here, the O/N ratio was 1.19 for the virgin membrane, which is a typical surface O/N ratio when compared with a serial of commercial RO membranes [17,33]. Interestingly, the O/N ratio instead followed a decreasing trend as more ODA@GO was added (for TFN-5, the O/N ratio approached ~1). Although the doped ODA@GO could be observed in the polyamide matrix (Figure 5h,i), the decrease of the O/N ratio on the membrane surface should not be attributed to the incorporation of ODA@GO. This is because the O/N ratio of the ODA@GO was ~3.7 (Table 2), a value significantly higher than that of the O/N ratio of polyamide. Rather, the decreasing O/N ratio reflects a higher crosslinking degree of the bulk PA nanofilm. A similar conclusion can be drawn from the analysis of zeta potential. As shown in Figure 8b, the TFC membrane was typically negatively charged in the pH range of about 4.2–9.7, due to the hydrolysis of the acyl groups to give the carboxyl groups on the membrane surface. After the doping of the ODA@GO nanosheets, the negative charge of the TFN RO membrane surface gradually lessened, which was likely caused by the smaller amount of free carboxyl groups, therefore implying a higher crosslinking degree [33].

### 3.3. The Effect of Confined Growth Mechanism on the Resultant PA Layer

As mentioned above, the ODA@GO doped TFN membranes have developed leaf-like nanostructure characterized surfaces, which are distinctive from the virgin TFC membrane. Especially at high ODA@GO concentration, ODA@GO nanosheets can be readily observed at locations where the leaf-like structures are observed (Figure 5h,i). Such a phenomenon can be explained by the confined growth of polyamide at the interface due to the limiting effect of the ODA@GO nanosheets to MPD diffusion. Specifically, the limiting effect can be interpreted in two ways: Firstly, the limited diffusion of MPD molecules into the organic phase results in higher MPD concentration at the interface [23], therefore resulting in a more intense IP reaction, hence resulting in a greater occurrence of the leaf-like structures [34]. Secondly, the presence of the ODA@GO nanosheets limit the growth of the nanobubbles in the z-direction (the direction that is perpendicular to the membrane surface), therefore the nanobubbles are more inclined to develop laterally and finally into leaf-like structures. Collectively, the decreasing trend of the apparent thickness, the increasing trend of the intrinsic thickness and the crosslinking degree of the PA layer agree well with the confined growth mechanism, as the apparent thickness is mainly governed by the vertical growth of the nanoscale structures, while the intrinsic thickness and crosslinking degree is mainly governed by the enhanced intensity of the IP reaction. The above-mentioned confined growth mechanism of the PA layer is illustrated in Figure 9.

### 3.4. Performance Evaluation of the As-Developed Membrane

As can be seen from Figure 10, with the increase in the incorporation amount in the membrane, water flux generally increased initially. For example, compared with the virgin membrane, the optimal brackish water flux increased by 11% to 47.9 L m^−2^ h^−1^ at a doping amount of 0.01% (*w*/*v*), and the salt rejection was at approximately the same level as the virgin membrane (~99.7%). The enhancement of the water flux can be attributed to the horizontal growth of the leaf-like structures, which enlarged the surface area of the polyamide nanofilm [17]. The increment of flux was also observed in the case of seawater desalination conditions. However, the optimal doping amount was 0.005% (*w*/*v*) when an optimum flux of 59.6 L m^−2^ h^−1^ was achieved, which had a 17% increase compared with the virgin membrane at the same testing condition. Simultaneously, the TFN-3 membrane achieved 99.6% salt rejection, a significant elevation from the ~99.1% for the virgin membrane. It even reached the rejection of some commercial seawater desalination membranes, and the flux is much higher than that of partial commercial membrane (Table 3). It is rather interesting to note that the ODA@GO TFN membranes had higher salt rejection than the virgin membrane at seawater operation conditions. This phenomenon can be explained by the solution diffusion mechanism [35,36]. For an ideally dense RO membrane, the solute flux is mainly dependent on the concentration difference between the feed and the permeate. Therefore, operating at high water flux (i.e., higher hydraulic pressure) helps to dilute the solute flux, resulting in lower salt concentration in the permeate. Therefore, the net result is higher salt rejection at higher operation pressure [11,12,17,21]. On the other hand, for the looser TFC membrane, operating at high pressure facilitates both the solute flux (convective diffusion) and the water flux. Hence, the net result was that its brackish water salt rejection (~99.7%) was significantly higher than its seawater salt rejection (~99.1%).

Meanwhile, the boron rejection was also enhanced by the doping of ODA@GO (Figure 11). The initial boron removal rate of the virgin TFC membrane was 59.2%. This value could be improved by 13.3% when an optimum ODA@GO amount was doped. Accompanying this, the coefficient of boron removal *B_s_* decreased significantly over the doping of ODA@GO. This decrease in the boron diffusion coefficient and elevation of boron rejection can be explained by the increased intrinsic thickness and the cross-linking degree of the PA nanofilm that increased the separation efficiency [17].

## 4. Conclusions

In conclusion, we have discovered in this study that the growth of the PA nanofilm under the confinement effect of the 2D ODA@GO nanosheets during the interfacial polymerization can effectively shape the nanoscale structures and customize the properties of the polyamide nanofilms. The ODA@GO nanosheets dispersed in the organic phase served as an effective barrier limiting the diffusion of amine molecules into the organic phase. As a result, the PA nanofilm was shaped with a significant amount of leaf-like structures, which promoted the horizontal growth of the PA nanofilm. As a net result, the apparent thickness of the PA layer was decreased but the overall effective surface area was enhanced, making the PA layer more efficient for water permeation. In the meantime, both the intrinsic thickness and the cross-linking degree of the PA nanofilm were enhanced due to the elevated amine concentration at the interface, rendering the PA nanofilm a better barrier for salt and neutral molecules such as boron acid. Therefore, we have demonstrated that proper doping of 2D nanosheets in the organic phase during IP reaction has the potential to produce more effective PA nanofilms. This interesting finding will pave the road for further studies to customize higher-selectivity PA-based polymeric TFN membranes for seawater desalination.

## Figures and Tables

**Figure 1 membranes-11-00488-f001:**
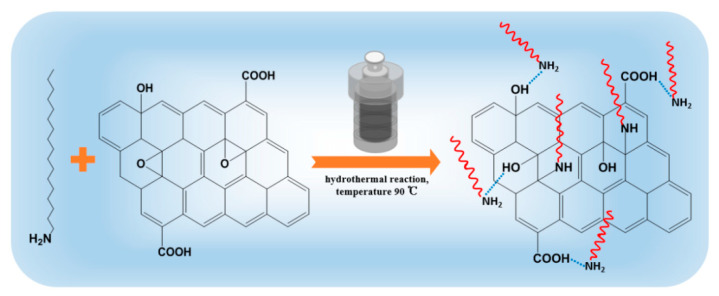
The fabrication process and mechanism of ODA@GO.

**Figure 2 membranes-11-00488-f002:**
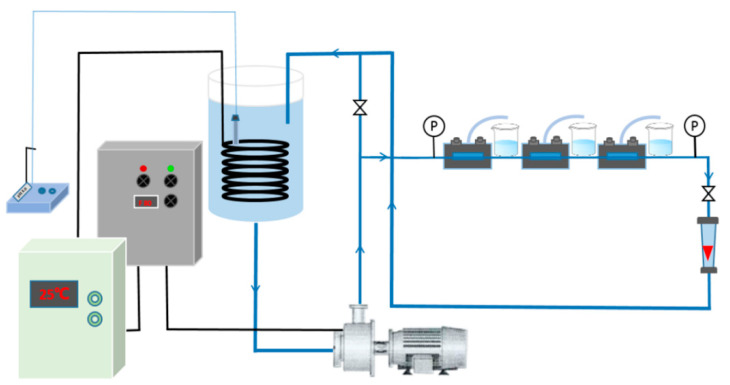
Schematic diagram of the experimental apparatus used to evaluate the permselectivity of the fabricated RO membranes.

**Figure 3 membranes-11-00488-f003:**
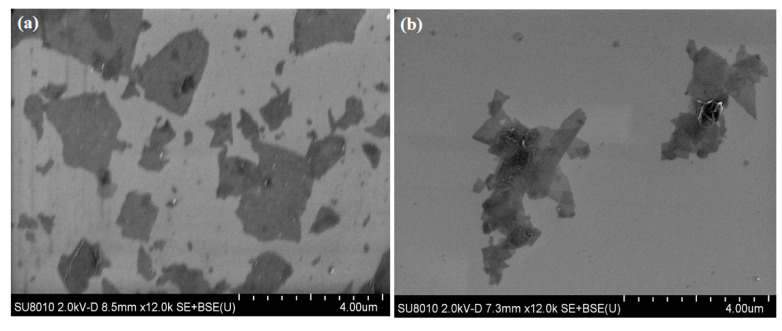
(**a**,**b**) SEM images of GO and ODA@GO, respectively; (**c**,**d**) AFM images of GO and ODA@GO, respectively.

**Figure 4 membranes-11-00488-f004:**
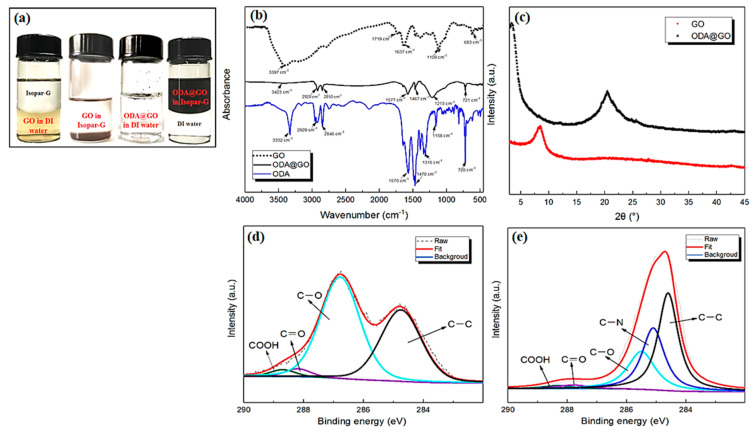
(**a**) Digital photos of GO and ODA@GO dispersion in solvents; (**b**) FTIR spectra of GO, ODA, and ODA@GO; (**c**) XRD analysis of GO and ODA@GO; (**d**,**e**) XPS-peak-differentiation-imitating analysis of GO and ODA@GO, respectively.

**Figure 5 membranes-11-00488-f005:**
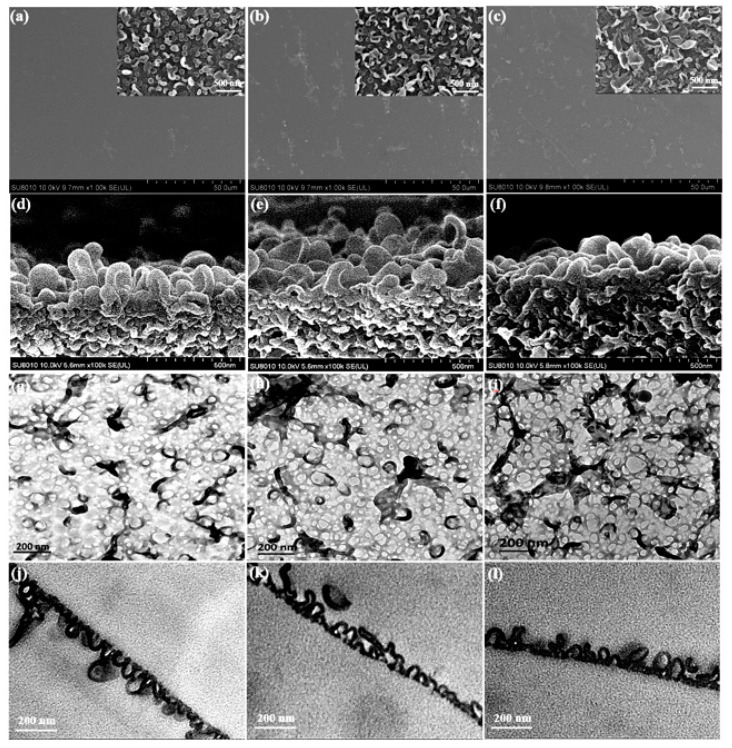
SEM and TEM micrographs of the top surface and cross-section for TFC and TFN membranes with different ODA@GO nanosheet loadings: (**a**,**d**,**g**,**j**) TFC (0%, *w*/*v*), (**b**,**e**,**h**,**k**) TFN—2 (0.003%, *w*/*v*) and (**c**,**f**,**i**,**l**) TFN—5 (0.02%, *w*/*v*).

**Figure 6 membranes-11-00488-f006:**
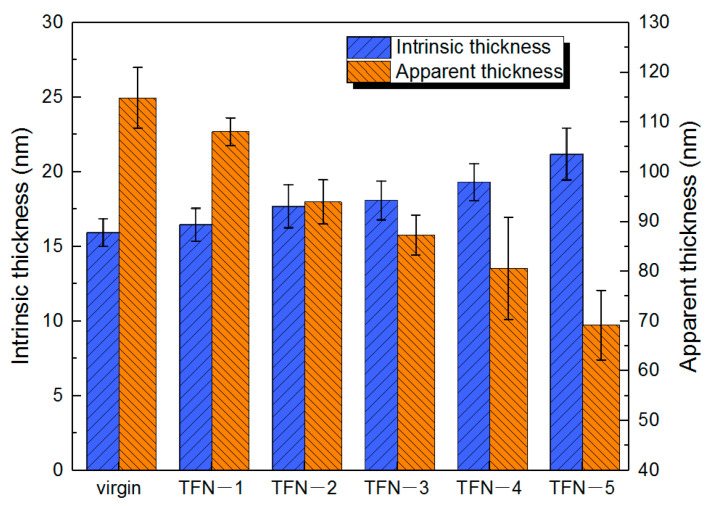
The comparison of intrinsic and apparent thickness in TEM images.

**Figure 7 membranes-11-00488-f007:**
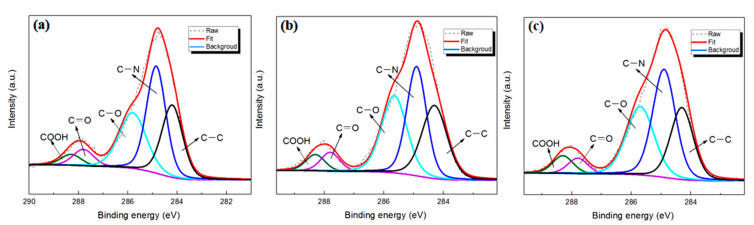
XPS peak analysis of the RO membrane surface: (**a**) Virgin; (**b**) TFN—2; (**c**) TFN—5.

**Figure 8 membranes-11-00488-f008:**
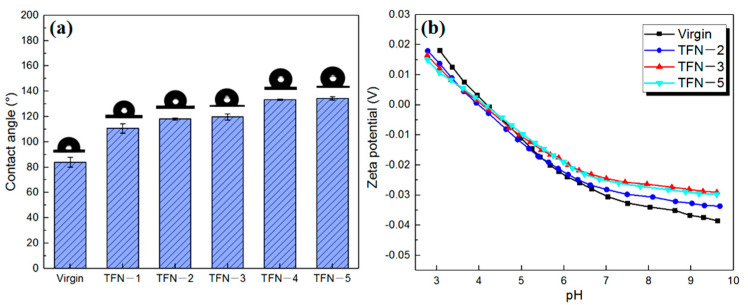
Contact angle results (**a**) and zeta potential analysis (**b**) of RO membranes.

**Figure 9 membranes-11-00488-f009:**
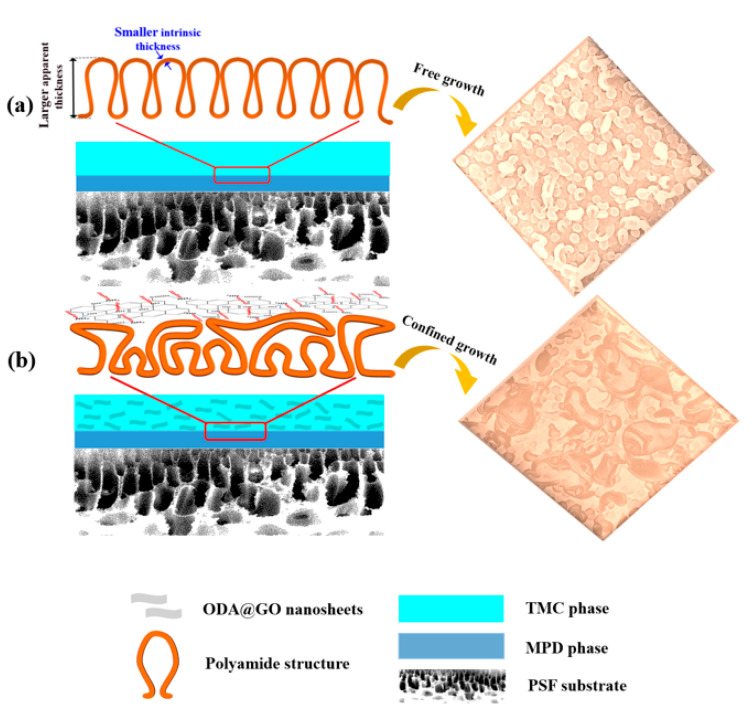
Schematic diagram of the confined growth of ODA@GO: (**a**) No ODA@GO, (**b**) ODA@GO was added as a barrier to constrain the preferred orientation of growth of the PA layer.

**Figure 10 membranes-11-00488-f010:**
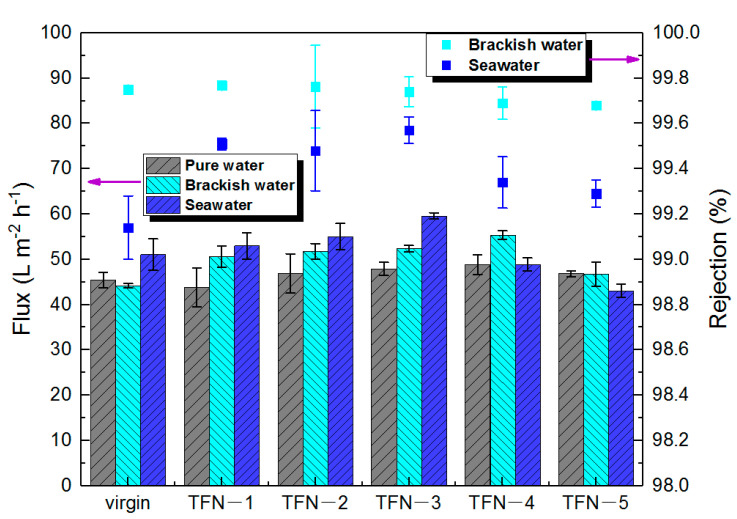
Effects of ODA@GO loading on the flux and salt rejections of the RO membranes under pure water, brackish water desalination, and seawater desalination tests.

**Figure 11 membranes-11-00488-f011:**
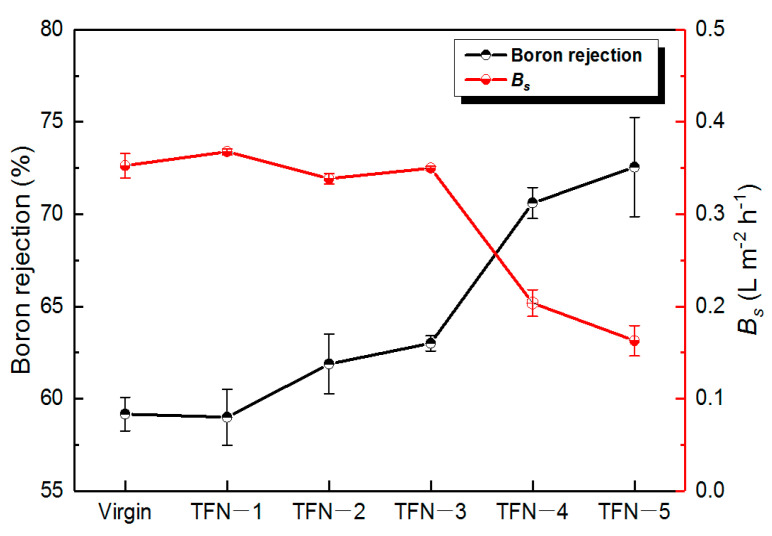
Rejection and permeability coefficients of 5 ppm boron through the RO membranes.

**Table 1 membranes-11-00488-t001:** Data comparison of the TFC and TFN RO membranes.

Sample	Intrinsic Thickness ^a^ (nm)	Apparent Thickness ^b^ (nm)	*R_a_* (nm)	Contact Angle ^c^ (°)	Zeta Potential ^d^ (mV)	*A* ^e^ (LMH bar^−1^)	*A*/*B* ^f^ (bar^−1^)	*B*′ ^g^ (LMH)
Virgin	15.93 ± 0.9	114.9 ± 6.1	47.3 ± 3.2	84.0	−34.94	2.84	25.81	0.46
TFN-2	17.69 ± 1.4	93.9 ± 4.4	40.1 ± 2.6	118.1	−31.63	2.93	26.63	0.27
TFN-3	18.09 ± 1.3	87.3 ± 3.9	40.0 ± 3.3	119.7	−26.36	2.99	26.46	0.24
TFN-5	21.19 ± 1.7	69.2 ± 7.0	33.5 ± 3.7	134.4	−27.09	2.93	23.90	0.37

^a^ Based on 10 measurements in TEM images of the wall thickness of PA; ^b^ Based on 10 measurements in TEM images of the integral thickness of PA; ^c^ Based on 6 measurements in static water contact angle of top surfaces of membranes; ^d^ pH = 8 during the membrane performance test; ^e^ Based on the pure water permeability coefficient; ^f^ Based on the water/NaCl selectivity; ^g^ Based on the 32,000 ppm NaCl permeability coefficient.

**Table 2 membranes-11-00488-t002:** Elemental composition of nanosheets and different RO membranes analyzed by XPS.

Sample	Atomic Percent (%)			Atomic Ratio		
	C	N	O	N/C	O/C	O/N
GO sheets	69.59	-	30.41	-	0.4369	-
ODA@GO sheets	88.11	2.53	9.36	0.0287	0.1062	3.6996
Virgin	74.74	11.55	13.7	0.1545	0.1833	1.1861
TFN-1	75.03	11.48	13.49	0.153	0.1798	1.175
TFN-3	76.58	10.98	12.44	0.1434	0.1624	1.133
TFN-4	76.79	11.07	12.13	0.1441	0.158	1.0958
TFN-5	77.77	11.13	11.09	0.1431	0.1426	0.9964

**Table 3 membranes-11-00488-t003:** Transport properties of RO membranes in this work and the SWRO commercial membrane at seawater desalination condition ^a^.

Sample	*J* (L m^−2^ h^−1^)	*R* (%)	*R_B_* ^b^ (%)
Virgin	51.1	99.1	59.2
TFN-2	55	99.5	61.9
TFN-3	59.6	99.6	63
TFN-4	53.9	99.3	70.6
SW30HR-380	27.4	99.6	91

^a^ 32,000 ppm NaCl with 5 ppm boron, an operation pressure ~55 bar, T = 25 °C and pH = 8; ^b^ Boron rejection.

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
