# Peer review of "Towards a High Rejection Desalination Membrane: The Confined Growth of Polyamide Nanofilm Induced by Alkyl-Capped Graphene Oxide"

_membranes, 2021, doi:10.3390/membranes11070488_

Round 1

Reviewer 1 Report

The paper entitled “Towards high rejection desalination membrane: the confined growth of polyamide nanofilm induced by alkylcapped graphene oxide” and written by Biqin Wu et al. is interesting. The problem is that I do not see coherence between the title and the topic of the manuscript and the results obtained. My recommendation is to reject this paper based on the following comments:

  1. Page 1, in the abstract. Authors wrote “The development of high rejection reverse osmosis membrane, especially at seawater desalination conditions, is the key to deliver high-quality product water with lower production costs”. Authors should take care, usually the RO membrane with higher rejection are the membrane with lower water permeability coefficient and therefore produce less water or require more feed pressure to be able to produce the same as other membranes. It means higher specific energy consumption so higher operating costs. The authors should remark in the introduction why it is necessary to manufacture membrane with higher solute rejection. This is not clear.
  2. Page 1, section introduction, third line. Authors wrote 5 cites in a row. I think it is too much. Taking into consideration that the references in this case are supporting a very general sentences with 2 o 3 cites is more than enough
  3. Page 2 line 1. Please clarify the issue about cost-effective according with my point one. Clarify why a membrane with higher rejection is more cost-effective. And again 5 cites in a row is too much for a sentence please, remove 2 or 3.
  4. Page 2, third paragraph. Write the meaning of MPD please
  5. Page 2, third paragraph a cite is missing after “Later, Yan et al.
  6. Page 5, the water permeability coefficient is a variable so please, write it in italics. Same with the solute permeability coefficient
  7. Page 5, last paragraph. Why the authors carried out the test for brackish water at such a high feed pressure (16 bar)?
  8. Page 5, last paragraph. The authors mentioned that delta p is the operating pressure but that is not true. According with the solution-diffusion equations, delta p is the net driving pressure and to calculate this term the authors have to take into consideration the osmotic pressure gradient, pressure drop (did the membrane has spacers?) in the feed channel and concentration polarization phenomena (this also has to be considered to estimate the solute permeability coefficient). Did the authors consider this?
  9. Please, write al the variables in italics.
  10. The authors also analyze the boron rejection however; in the introduction the authors did not mention anything about boron rejection problem in seawater as one of the main reason to elaborate new membranes with higher rejection. Please, write a few lines about boron rejection by reverse osmosis incorporating a few cites. here I suggest some:
    1. Practical considerations of wastewater–seawater integrated reverse osmosis: Design constraint by boron removal
    2. Different boron rejection behavior in two RO membranes installed in the same full-scale SWRO desalination plant
    3. Boron removal with modified polyamide RO modules by cross-linked glutaric dialdehyde grafting
    4. Study of the influence of temperature on boron concentration estimation in desalinated seawater for agricultural irrigation
    5. Comparison analysis of different technologies for the removal of boron from seawater: A review
    6. High boron removal polyamide reverse osmosis membranes by swelling induced embedding of a sulfonyl molecular plug
    7. Comparative performance of FO-RO hybrid and two-pass SWRO desalination processes: Boron removal
    8. ‘High-pH softening pretreatment’ for boron removal in inland desalination systems
  11. Page 6, end of first paragraph too many citations in a row. IT is not appropriate, if the authors want to take into consideration those paper it should be commented separately.
  12. Page 15, Table 3. Considering the boron rejection rates, the membranes modified by the authors (TFN-2, TFN-3 and TFN-4) could not be in operation in real SWRO systems. Even in real operation the membrane SW30HR-380 had problems to reject boron to reach a 1 ppm in the permeate. And reject NaCl it is not a big issue nowadays considering the characteristics of the commercial SWRO membranes. One more thing that should be commented by the authors is the stability of the “new membranes” in long-term operation which has not been tested.
  13. According with my point 12, It is hard to understand the purpose of this study, first the authors wanted to make a high selectivity membrane but, the table 3 does not show a high selectivity membrane rather a high flux membrane. So, I do not see the relation between the obtained results and the title and introduction of the manuscript. Please, could the authors clarify this point.
  14. The introduction should be extended and there are missing references that are relevant and related with the topic:
    1. A New Method for a Polyethersulfone-Based Dopamine-Graphene (xGnP-DA/PES) Nanocomposite Membrane in Low/Ultra-Low Pressure Reverse Osmosis (L/ULPRO) Desalination
    2. Can graphene and graphene oxide materials revolutionise desalination processes?
    3. Effects of the Substrate on Interfacial Polymerization: Tuning the Hydrophobicity via Polyelectrolyte Deposition
    4. Functional materials in desalination: a review
    5. GO/PVA-integrated TFN RO membrane: Exploring the effect of orientation switching between PA and GO/PVA and evaluating the GO loading impact
    6. Tuning the Surface Structure of Polyamide Membranes Using Porous Carbon Nitride Nanoparticles for High-Performance Seawater Desalination
    7. Doping MIL-101(Cr)@GO in polyamide nanocomposite membranes with improved water flux
    8. Surface modification of reverse osmosis membranes by grafting of polyamidoamine dendrimer containing graphene oxide nanosheets for desalination improvement
    9. Role of Cellulose Micro and Nano Crystals in Thin Film and Support Layer of Nanocomposite Membranes for Brackish Water Desalination
    10. Etc….

Reviewer 2 Report

The authors discuss the PA surface structure and neutral molecules rejection alternation after incorporating ODA-GO. It is a complete investigation. However, I do have a couple of comments and questions. I suggest the submitted manuscript a major revision before publishing in Membranes. 

  1. About the experimental setup, could the author provide the flow rate and surface cross-flow velocity? The active membrane area 19.63cm2 is for a single module or a sum of three of them? The membrane module is cycle or rectangle? A rectangle will be more approaching to the real flow status.
  2. Authors conclude the 2D ODA-GO has the potential to produce a more effective PA layer. Could the author explain in more detail how good is the 2D shape benefit the PA layer? and compare to the 3D ball shape or hollow fiber shape.
  3. Whether the additive (ODA-GO) still has a high possibility to occur defect between the PA polymer chain length and additive?

Reviewer 3 Report

Authors proposed to increase the performance of polyamide TFC membranes aimed for Reverse osmosis, by inclusion of Graphene oxide during polymerization step. The experiments have been adequately performed and designed including a lot of characterization techniques to assess results. Unfortunately these results are  not very promising, at least in terms of water desalination.

English must be improved with some phrasing and wording difficult to understand.

Some minor comments follow:

  • Abstract: First phrasing (from “The development…” to “…In this paper”) is more introduction and makes no sense to include in abstract.
  • I don’t understand the use of levy in this context (see translator comment)
  • Introduction, top: surely TFC membranes are the most widely use (certainly we can include the original Loeb-Sourirajan cellulose acetate RO membrane as a TFC) but polyamide ones cannot be assured as the most preferred ones. In any case, reference to this assert should include some review on RO desalination comparing different membrane types and materials.
  • Page 4, top: Fourier is a name, so caps should be used (French mathematician)
  • Page 4, top: pallet? Do you refer to a tablet or disk?
  • Section 2.5: Eliminate “ Among these,” so placed has no meaning or rewrite to clarify
  • Please define apparent and intrinsic thickness, is the first time I see this terms referred to membranes. Do you mean overall thickness and active layer or support one? Please explain graphically
  • Section 3.1: It is not clear how do you measure nano sheets size, probably image analysis or line analysis for AFM, but please explain it
  • Figure 4a: authors extract a lot of conclusions about material behaviour from a not so clear picture
  • Figs 5a-c: the only interest of these figures is in the inserts which by the way are not explained.
  • Figs 5g-i: I don’t know which adds TEM to the SEM pictures. Possibly they are used to measure the intrinsic/apparent thickness but then SEM cross section are redundant
  • Similarly Figures of AFM pictures do not add any insight, just only the roughness parameters which in fact is not tabulated could be of some interest
  • Table 1: based on 10 measurements…. (typo). On the other side it should be necessary to include a measurement of both intrinsic and apparent thickness in some of the pictures to know what are you exactly talking about
  • 11 and 122: results for boron rejection are somehow promising to justify the work but those for water give not too much advance, for example change in flux is same order of experimental error. On the other side you have not defined brackish and seawater (composition, conditions,…), which rejection are you talking about? Sodium chloride, any salt, or what?
  • Too much references, include only those necessary

Round 2

Reviewer 1 Report

The paper has been improve considerably. The authors have adressed all my comments

Author Response

Dear reviewer,

    Thanks for the reviewer's affirmation and reply.

Yours sincerely,

Song xiaoxiao

Reviewer 2 Report

 accept

Author Response

(The authors gave the same response as above.)

Reviewer 3 Report

See my comments in attached answer to your response

Author Response

Dear Reviewer:

    We have modified and replied to the comments and questions raised above. Please refer to the attachment for details. Thank you.

Yours sincerely,

Song xiaoxiao  
